# An Overview on Dietary Polyphenols and Their Biopharmaceutical Classification System (BCS)

**DOI:** 10.3390/ijms22115514

**Published:** 2021-05-24

**Authors:** Francesca Truzzi, Camilla Tibaldi, Yanxin Zhang, Giovanni Dinelli, Eros D′Amen

**Affiliations:** Department of Agricultural and Food Sciences, Alma Mater Studiorum-University di Bologna, 40127 Bologna, Italy; camilla.tibaldi2@unibo.it (C.T.); yanxin.zhang2@unibo.it (Y.Z.); giovanni.dinelli@unibo.it (G.D.); eros.damen2@unibo.it (E.D.)

**Keywords:** polyphenols, metabolism, bioavailability, LogP, biopharmaceutical classification system

## Abstract

Polyphenols are natural organic compounds produced by plants, acting as antioxidants by reacting with ROS. These compounds are widely consumed in daily diet and many studies report several benefits to human health thanks to their bioavailability in humans. However, the digestion process of phenolic compounds is still not completely clear. Moreover, bioavailability is dependent on the metabolic phase of these compounds. The LogP value can be managed as a simplified measure of the lipophilicity of a substance ingested within the human body, which affects resultant absorption. The biopharmaceutical classification system (BCS), a method used to classify drugs intended for gastrointestinal absorption, correlates the solubility and permeability of the drug with both the rate and extent of oral absorption. BCS may be helpful to measure the bioactive constituents of foods, such as polyphenols, in order to understand their nutraceutical potential. There are many literature studies that focus on permeability, absorption, and bioavailability of polyphenols and their resultant metabolic byproducts, but there is still confusion about their respective LogP values and BCS classification. This review will provide an overview of the information regarding 10 dietarypolyphenols (ferulic acid, chlorogenic acid, rutin, quercetin, apigenin, cirsimaritin, daidzein, resveratrol, ellagic acid, and curcumin) and their association with the BCS classification.

## 1. Introduction

Phenolic compounds (PCs) are secondary plant metabolites, characterized by an aromatic ring and several attached hydroxyl groups. These compounds offer protection to the plant from pathogens, free oxygen radicals, UV rays, and parasites [1].

Polyphenols represent a large and varied group of at least 10,000 known different compounds that could be unified by the presence of one or more aromatic rings with one or more hydroxyl groups in their chemical structure [2,3]. For some plant products, for example, some exotic fruits or cereals, the composition of polyphenols is still poorly known [4]. Regarding dietary PC, the current known compounds are about 8000 variants and they are naturally found in common fruits, vegetables, and beverages and, according to the number of phenolic rings they contain, they could be classified into main four subclasses: [5] flavonoids, including flavonols, flavones, isoflavones, flavanones, anthocyanidins, and flavanols; phenolic acids subclass, which is divided between those compounds derived from hydroxybenzoic acids, such as gallic acid, and those derived from hydroxycinnamic acid, like caffeic, ferulic, and coumaric acid; and stilbenes and lignans (Figure 1) [6,7]. In addition to these, there are other subclasses that are not included among the currently known, i.e., alkylphenols, curcuminoids, furanocumarins, hydroxybenzaldehydes, hydroxybenzoketones, tyrosols, and so on [8].

PCs are derived from a common biosynthetic pathway, involving precursors from the shikimate and/or the acetate–malonate pathways [9]. In addition, their role in the prevention and improvement of human health has been widely demonstrated. There is growing evidence to support antioxidant, anti-inflammatory, anti-tumoral, and anti-cardiovascular disease roles attributable to polyphenols [10,11,12,13,14,15,16,17,18]. There are numerous different types of PCs, all characterized by different chemical structures that have distinct properties [19]. It was reported that the specificity of the health benefits, conferred by a single PC, is based on specific chemical classes [20].

As PCs are bioactive compounds, it is important to study their distribution within the human body. More specifically, in order to express their therapeutic effect, PCs should undergo pharmacological metabolism reactions, with consequent conversion into more soluble metabolites, and subsequent excretion [21,22]. From this perspective, it is important to study the health potential of PCs, and this may be done through the biopharmaceutical classification system (BCS). This classification evaluates the capacity of drugs (and thus also bioactive compounds) to pass through lipidic biological membranes as well as to interact with aqueous solutions during their metabolism within the human body. Hence, BCS allows to characterize the health capacities of a PC.

The aim of this work is to analyze the nutraceutical potential of 10 dietary PCs, in terms of intestinal absorption, permeability, solubility, and BCS classification of these compounds, and to compare them with the literature currently available, in order to establish an association between all these factors. Furthermore, it was chosen to focus this work only on 10 dietary PCs because they belong to most of the subfamilies of known polyphenols, and thus can be representative for an overall evaluation of the beneficial prospective of dietary PC present in nature. Besides, in the current literature, there is enough and exhaustive information on permeability, solubility, and especially BCS classification only on these PCs, univocally directing the discussion of this review to them.

## 2. Polyphenol Metabolism

### 2.1. Metabolism Course

In mammals, PCs are subject to both phases I and II of drug pharmacokinetic metabolism, respectively, as represented in Figure 2. PCs are ingested mainly in a conjugated form, as O-glycosides (step 1, Figure 2). Metabolism of glycosylated PC is initiated in the oral cavity [22], after contact with the glycosidase enzymes of oral microflora, as demonstrated by Kamonpatana et al. [23] for anthocyanidins. However, most PCs continue intact along the digestive tract [22]. On arrival in the stomach and inside the small intestine mucosa, the glycosides are converted by a hydroxylation reaction into their corresponding aglycones (phase I drug metabolism) (step 2, Figure 2). This reaction is assisted by β-glucosidase enzymes expressed by the intestinal microbiota. In this way, aglycones may pass from the gut lumen to the cytosol of the enterocytes, predominantly by passive diffusion (step 3, Figure 2), or by protein carriers, such as P-glycoprotein (P-gp) and co-transporters for sodium-glucose transporter (SGLT1) [24,25,26]. Some hydroxycinnamic acids, such as ellagitannins, are resistant to enzymatic digestion in the small intestine and, therefore, pass directly to the colon, where they are metabolized by microbiota into aglycones [27]. Once the aglycones have been absorbed into the enterocytes or colon cells, they move through the portal vein (step 4, Figure 2) to the liver, where they are further conjugated (phase II drug metabolism) to become O-glucuronides and O-sulphates (step 5, Figure 2). A variable portion of the phenolic conjugates is then excreted into the bile and re-enters the small intestine to undergo the metabolic cycle once again [28,29,30]. Finally, the resultant phenolic conjugates (O-glucoronides/O-sulphates) are transported to the bloodstream by plasma proteins until they are excreted in urine [31,32,33].

### 2.2. PC Metabolites Have Different Biological Activities

Biotransformation, during the metabolic processes in humans, is determined by the structural characteristics of the specific phenolic compound [9]. This is because the chemical structure of the compound is specific in promoting the action of only selected intestinal enzymes and gut microbiota species. It has been shown that gut microbiota are involved in the release of phenolic aglycones and hepatic O-glucuronides [28,34]. According to the different structural subfamilies, PCs undergo intestinal bio-transformations by specific microbiota families [31,35,36,37,38,39,40]. Furthermore, it was observed that each single phenolic compound metabolized generates numerous metabolite byproducts, usually two or three, but also many more, i.e., glycosylated quercetin produces up to 20 metabolites. According to Del Rio [41], all these modifications during absorption have a profound influence on the biological activity of the resultant phenolic metabolites, as these may play an active role within different pathways in the human body. An example is the activation of the transcription factor nuclear factor (erythroid-derived 2)-like 2 (Nrf-2) [42]. It was shown that protocatechuic acid, a metabolite from anthocyanins, is a known Nrf2 activator [43,44]. Likewise, caffeic acid metabolites have Nrf2 activating properties [45,46,47]. In addition, another protective function pathway was found to be influenced by phenolic metabolites; it was shown that methylated scutellare presents an inhibitory effect on H_2_O_2_-induced cytotoxicity in PC12 cells, thus indicating protective activity [48]. In order to be able to perform all the metabolic reactions, it is necessary that PC, as well as all xenobiotics, administered orally, enter the intestinal epithelium to reach the blood and lymphatic circulation [49]. The transcellular mechanisms required to permit the entry of these compounds into the intestinal mucosa are as follows: passive diffusion, carrier-mediated active facilitated transport, and paracellular transit in tight junctions [50,51]. Nevertheless, generally, the majority of drugs enter the cells by passive diffusion [52]. PCs, characterized by low molecular weights, and that are sufficiently hydrophobic and non-charged, are permitted to be transported by passive diffusion [24]. This involves the production of biliary salts and the formation of micelles, which permeate through the translocation of the apical membrane of the enterocytes [53]. Some PCs, such as hydroxytyrosol, tyrosol, p-cumaric acid, apigenin, and luteolin, are selectively combined in micelles and absorbed differently [54]. In addition, in an in vitro study using Caco-2 cells to test the permeability of six dietary polyphenols (caffeic acid, chrysine, gallic acid, quercetin, resveratrol, and rutin), it was shown that several chemical-physical features are related to the passive diffusion transport capability of molecules through cells. These factors include the lipophilicity (expressed as partition coefficient logarithm, LogP), molecular weight, ionization state the number of rotatable bonds (RB), and number of hydrogen-bonding acceptor/donor (HBA/HBD), respectively [55]. Moreover, it was observed that the scarcely lipophilic ferulic acid (FA) passes through transcellular transport, by tight junction [56]. Regarding structurally complex PCs, such as gallotannins and ellagitannins, hydrolysis processes permit conversion into smaller molecules, thereby facilitating assimilation in simpler forms by enterocytes. However, this conversion reaction cannot occur in the small intestine, so they pass directly into the large intestine where they are fermented by the microbiota and then can be absorbed by passive diffusion at the level of the colon [9,57,58]. All these transport mechanisms influence the bioavailability of these compounds in humans [49].

## 3. Polyphenol Bioavailability

Bioavailability is a term used in pharmacokinetic language to indicate the fraction of the drug that reaches the systemic circulation without any chemical modification. Recently, it has also been used for food nutrients. According to the current accepted definition, bioavailability is the proportion of the nutrient that is digested, absorbed, and metabolized through normal pathways [54].

Each class of PC has its own unique chemical structure that results in specific solubility and lipophilicity, which in turn affects the bioavailability. These parameters influence both the rate and degree of absorption of phenolic metabolites in the human body during metabolism, before being incorporated into plasma circulation [59]. In the studies of Marrugat [60], Fito [61], and Tian [62], the bioavailability of various PC was analyzed by detecting plasma and urine concentrations after the ingestion of pure compounds or food with a known phenolic content. It was found that phenolic concentrations in either plasma or urine were not directly related to the respective concentrations in target tissues, but were instead dependent on their metabolic form. This is explained by the fact that PC, although ingested in a glycosylated form, undergo metabolic processes, such as hydrolysis/hydroxylation by the intestinal enzymes and subsequent conjugation with either glucuronic or sulfonic acid in the liver. This leads to a change in their chemical structure, thereby modifying the respective lipophilicity and solubility characteristics, permitting entry into the blood circulation. Therefore, PCs are found in the blood circulation in conjugated forms. To confirm this, quercetin and daidzein aglycones were not found in either plasma or urine after they are ingested, but in their conjugated form (after phase II metabolism) [63]. Resveratrol represents an exception, as it undergoes glycosylation in order to protect the compound from oxidative degradation. Hence, the glycosylate resveratrol is chemically more stable and soluble, and consequently more easily absorbed in the human gastrointestinal tract [64]. Finally, it was shown that large size conjugated metabolites are eliminated in bile, while small conjugates, such as monosulphates, are preferably excreted in urine [59].

In this perspective, BCS classification may allow prediction of health effects for PC contained in foods and thus are administered mostly orally. In 2000, the Food and Drug Administration (FDA) purposed the BCS system as an approach to avoid in vivo tests when drugs are also characterized by rapid dissolution [65]. More in particular, it was highlighted that high permeability of a drug does not limit the absorption of a compound during its transit in the intestinal system. In addition, high solubility of a drug will not limit its dissolution and consequently neither its absorption, thus the gastric emptying process is the only limiting step for the absorption of a highly soluble and highly permeable compound [66]. However, some studies have shown that the FDA’s BCS guidance, despite supporting studies on bioavailability through in vitro test such as on Caco-2 [66], may not always be enough to correctly predict the extent of drug absorption in humans [67,68].

To study the bioavailability of an active compound, the most reliable measure is the area under the plasma drug concentration curve versus time (AUC). AUC is directly proportional to the total amount of unchanged drug that reaches systemic circulation. Plasma drug concentration increases with the extent of absorption, and the maximum plasma concentration is reached when the drug elimination rate equals the absorption rate. The time peak, which is reached when maximum plasma drug concentration occurs, is the most widely used general index of absorption rate; the slower the absorption, the later the peak time. To determine significant AUC values, a cutoff of 80% was defined [69]. For dietary supplements, herbs, and other nutrients, such as PC, in which the administration is nearly always oral, bioavailability can be identified as the quantity or fraction of the ingested dose that is absorbed [70]. Hence, in this review, the maximum plasma concentration and time peak of the PC examined will be taken into consideration for the evaluation of their bioavailability.

### 3.1. Hydroxycinnamic Acids

In the early study of Lempereur [71], it was seen that the source of hydroxicinnamic acids in foods is relatively varied. For instance, FA is the most abundant phenolic acid found in cereal grains and represents up to 90% of the total polyphenol content. In the study of Zhao [72], it was shown that FA has a very high bioavailability in rats [73,74]. It was observed that FA, after undergoing absorption by intestinal epithelial cells and conjugation reactions [75], was present in both plasma and urine mainly in its conjugated form [76]. In several clinical studies carried out on rats, FA administrated at 8 μmol kg^−1^ [72], and between 6 and 15 mg kg^−1^ [77,78,79], displays high plasma concentration between 8174.55 ng L^−1^ and 0.444 mg L^−1^. Higher results were detected with a higher dosage (20 mg kg^−1^) [80], with a peak plasma concentration of around 12 mg L^−1^, whereas it seems that even higher doses (0.5 and 1.5 g kg^−1^) lead to a maximum plasma concentration of around 12 ng mL^−1^, showing a limiting rate of absorption. Furthermore, FA time peak was observed at less than 1 h, which implied that it is rapidly absorbed in rat plasma after oral administration [81].

Caffeic acid is the most representative hydroxycinnamic acid present in nature, and can be found in food (mostly fruits) as well as in its ester form, as chlorogenic acid [19]. Clinical studies observed that chlorogenic acid is rapidly absorbed and metabolized by the intestine, thus the digestion metabolites may be detected in plasma between the first 5 min and 1 h after ingestion in rats [82,83]. The current literature showed that chlorogenic acid absorption is dependent on administered doses, i.e., a low administered dose (1–100 mg kg^−1^) [83,84,85] leads to a concentration between 0.55 and 91 ng mL^−1^ of caffeic acid in plasma concentration; likewise, higher doses (400 mg kg^−1^ from *Lonicerae Japonicae Flos* extracts) [82], for which peak plasma concentration was registered at around 1500 mg mL^−1^, show an absorption rate dependent on the ingested dose. Regarding urinary excretion, the concentration of chlorogenic acid found after 24 h from ingestion was around 30–34%, indicating that this compound is rapidly eliminated as well as it is rapidly absorbed [83].

### 3.2. Flavonols

The most studied flavonols are rutin and quercetin, and they are mostly found in buckwheat, asparagus, and citrus fruits, but also in peaches, apples, and green tea [86,87]. In pharmacokinetics studies carried out on rats and human volunteers [88,89,90,91,92], low bioavailability of rutin was shown, due to its hydrophilic nature, thus suggesting that it cannot diffuse easily through cell plasma membranes. To be absorbable, rutin needs to undergo to hydroxylation into quercetin. In fact, after oral administration of rutin (328 μmol kg^−1^), only quercetin sulfates and glucuronides were detected in serum, with a concentration of 2 and 5 nmol mL^−1^ [89]. This evidence was in line with previously observations, where, after oral rutin administration (500 mg), the absorption rate was 40–200 ng mL^−1^ of quercetin [88] and showing the metabolism changes that occur on this compound, which are necessary to absorb it within the human body.

Quercetin bioavailability was found to be very poor, as it is rapidly metabolized in the human body; therefore, in the conjugated form (quercetin metabolite), its beneficial capacities are limited compared with the aglycon form [91]. Furthermore, the total quercetin conjugates measured in plasma concentrations after oral ingestion are very low. In the study of Dong [90], rats showed rapid absorption of quercetin (8.51 mg kg^−1^) from *Matricaria chamomilla L.* extract, with a final plasma concentration around 0.29 µg mL^−1^ detected after 0.79 h (47 min) from ingestion, confirming that it is rapidly absorbed after oral administration. In Graefe study [91], human volunteers who ingested 100 mg of quercetin glycosides showed a maximum plasma concentration at 2.12 µg mL^−1^ and urine concentration at 4.5%. This evidence confirmed the rapid absorption of quercetin aglycone, which is supposed to occur in the upper part of the intestine, thus involving active absorption mechanisms. Kaşıkcı [93] demonstrated that absolute bioavailability (2.01 μM) of quercetin was attained after ingesting the compound suspended in aqueous solution.

### 3.3. Flavones

Regarding flavones, celery, red peppers, chamomile, mint, parsley, rosemary, oregano, traditional Chinese herbs, and ginkgo biloba are the major sources of this subclass [94]. It was shown that apigenin administered at both low doses (13.82 mg kg^−1^) [90] and high doses (60 mg kg^−1^ and 100 mg kg^−1^) [95,96], is similarly absorbed, with max plasma concentration registered between 0.14 and 1.33 μg mL^−1^. Moreover, in the study carried out on six men, the apigenin conjugates detected in urine after 24 h from the administration lead to an excretion rate around 0.22%, suggesting that most of the apigenin ingested is rapidly metabolized or is excreted unabsorbed, thus showing high permeability of this PC within the human body [97].

Following the flavones subclass, cirsimaritin, mostly found in rosemary and oregano, was also investigated. In a pharmacokinetic study on rats [98], 8 mg kg^−1^ of cirsimarin (glycosyde form) was administered from crude extract of *Microtea debilis* and, 5 h after ingestion, the low permeability of this PC was demonstrated: the determination of plasma concentrations of cirsimaritin aglycone was 0.138 µM and the urine concentrations after 5 h from oral administration were only 5.05 µM (3–5%). These results showed that cirsimarin is not absorbed from the gastrointestinal tract, but in the stomach, and then it must be converted to cirsimaritin to produce systemic healthy effects within the human body.

### 3.4. Isoflavones

Isoflavones are present almost exclusively in leguminous plants, particularly daidzein, which is found in large quantities in soybeans and soymilk [99]. Based on the several studies carried out on rats or human volunteers, daidzein showed low bioavailability. In fact, in experiments with both low concentration (from 0.4 to 1 mg kg^−1^) [100,101] and high concentration of oral administered daidzein (from 30 to 50 mg kg^−1^ and 418 μmol L^−1^) [102,103,104], the serum peak registered (173.1 ng mL^−1^ and from 0.38 to 2.5 μmol L^−1^) appeared within 2 or 8 h from ingestion [102], highlighting its rapid absorption from the gastrointestinal tract, and indicating the low bioavailability of this compound.

### 3.5. Stilbenes

Despite the low quantities of stilbenes in the human diet, resveratrol is both the most representative polyphenol and the widely studied, as it is considered the main one responsible for health benefits [105,106,107]. More in particular, it was indicated by numerous recent studies that resveratrol presents several benefits to human health, such as antibacterial, antioxidant, anti-inflammatory, and anticancer activities [108,109,110,111]. Resveratrol has been detected in numerous plants, particularly in red grapes, and thus is highly concentrated in red wine and grape juice [112]. Regarding bioavailability of this compound, after oral administration, resveratrol is absorbed by passive diffusion or by membrane transporters within the intestine, where it undergoes metabolic reactions, and then the resulting metabolite is released in the bloodstream, where it can be detected [113]. It was demonstrated from several clinical studies that, after oral dose of resveratrol between 25 and 150 mg [114,115,116,117,118,119], the max plasma concentration registered ranges between 491 ng mL^−1^ and 471 μg L^−1^. Moreover, lower doses of resveratrol were studied (0.5 and 1 mg) [120] and a low plasma concentration of the glucuronidated form at 130.19 ng mL^−1^ was registered, whereas for higher doses (from 500 mg to 5 g) [119,120,121,122], the plasma concentration detected was between 0.5 μg mL^−1^ and 4 μg mL^−1^. These several studies demonstrated that the absorption rate is dependent on the orally administered dose of resveratrol. Urinary excretion of this PC and its metabolites was rapid, with 77% of all urinary agent-derived species excreted within 4 h after the lowest dose [121]. As previously explained, when resveratrol is taken orally, it is metabolized to its glycosylated form, which increases its stability and solubility, allowing this compound to be more easily absorbed [64]. Hence, it was concluded that the systemic bioavailability of resveratrol is high, showing high permeability within the human body, thus accumulation of potentially active resveratrol metabolites may produce healthy effects within the human body [115].

### 3.6. Tannins

Ellagic acid is a natural phenolic antioxidant found in many fruits and vegetables, such as walnuts, pecans, cranberries, raspberries, strawberries, grapes, peaches, and pomegranates. Clinical studies carried out on this compound showed its low bioavailability. More in particular, both low oral administration (20–25 mg) [123,124] and high oral administration (85.3 mg kg^−1^ and >500 mg) [125,126] showed a plasma concentration in human patients between 30 and 200 ng mL^−1^. An exception can be made for the 40 mg dose, which can be considered halfway between a low and high administered dose. The serum peak concentration was registered around 200 ng mL^−1^, which was similar to the serum peak concentration obtained from a high dose of ellagic acid (>500 mg). These results showed that the absorption system of ellagic acid becomes saturated above a certain dosage of this PC and, therefore, the maximum detectable plasmatic concentration seems to reach its plateau, which appears to be around 200 ng mL^−1^ in the case of ellagic acid. Moreover, this peak concentration in the serum was reached after 1 h from the administration, owing to the fact that 50% of total ellagic acid was shown to bind to blood proteins after intestinal absorption (within the first 30 min after oral administration) [127].

### 3.7. Curcuminoids

Among Curcuminoids, curcumin covers relevance in Southeast Asia, as it is abundantly found in turmeric, which is a spice widely used in Southeastern Asian countries’ culinary traditions [128]. It is relevant to note the highly lipophilic nature of curcumin (high LogP), attributable to the methine-rich segments that connect the polar regions [129]; this is also reflected in its capacity to interact with biomembranes [130,131]. However, its therapeutic potential is still debated owing to its poor bioavailability in humans. Curcumin has been shown to have low permeability and to be poorly absorbed from the small intestine, whereas conjugative metabolism in the liver is extensive [129]. The low availability is also caused by the binding of curcumin to enterocyte proteins, altering its chemical structure [132]. It was seen that curcumin is ineffectively transported through the intestinal mucosa into circulation [133]; this molecule can undergo biotransformation within the intestinal mucosa or directly in the bloodstream [134,135].

Pharmacokinetic studies demonstrated that a high dosage of this compound, between 8 and 12 g daily [128,136,137,138,139], displayed a low plasma concentration, ranging between 50 ng mL^−1^ [139,140] and 2 μg mL^−1^ [128], and similar with low dosages (from 100 mg to 4 g daily) [136,137,140,141,142,143,144,145,146], with peak plasma concentrations registered around 0.51 nM [136,143], 15.8 nmol L^−1^ [140], and 12.2 and 96 ng mL^−1^ [144,145,146]. In both studies of Mahale [143] and Dhillon [139], the maximum plasma concentration was detected 2 h after ingestion of curcumin and urinary levels collected after 24 h were 210 and 510 nmol L^−1^ of curcumin glucuronides [140], showing that its metabolites have a short time period within the human body, meaning curcumin is not expected to be able to exert its beneficial health effects.

## 4. Permeability and Solubility of Polyphenols

BCS classifications of drugs bioavailability are based on their assigned value of permeability and solubility; if, by one side, the definition of solubility is clear and well established, the permeability concept needs some insights. Apart from active mechanism of trans-membrane selective transport, passive diffusion is the fundamental mechanism for the entry of a drug into human cells, and thus the resulting functional performance. From the perspective of studying the permeability of an orally administered bioactive compound, its diffusion capacity within the human organism tissues is the key property. For this reason, it is fundamental to evaluate the capacity of these compounds to pass through lipidic biological membranes as well as to interact with aqueous solutions (such as gastric acids and bile acids in the gastrointestinal tract). Measurable chemical-physical parameters of a given molecule that can be easily associated with diffusional processes in those environments are aqueous solubility and lipophilicity. Generally, the most lipophilic molecules show scarce water solubility [52]; however, it is necessary that a potentially bioactive molecule shows a specific balance between water solubility and lipophilicity to be biodisposable. A useful measure to define this equilibrium is the partition coefficient, or its logarithmic expression, LogP.

### 4.1. LogP

Even if lipophilicity of a specific compound is a function of several independent parameters (i.e., molecular mass, non-polar surfaces, total number of H-bond donor and acceptor, ionizable groups, and so on), it is commonly known that the LogP value can be an easy measure to express and compare this property [147]. The LogP value is the base 10 logarithm of the ratio called the partition coefficient, the ratio between the concentrations at the equilibrium of a substance dissolved in a biphasic system 1-octanol/water. The higher the LogP value, the more lipophilic the molecule; conversely, LogP values smaller than 1 are related to hydrophilic behavior. It is relevant to underline that to state that the LogP value is directly related to lipophilicity is a strong simplification, and some more accurate descriptors have been developed to better describe molecular features. For example, LogD (distribution coefficient logarithm) is basically the same concept as LogP, but it takes into account the ionization capability of some molecules over the whole pH scale. LogD is formally a pH-dependant LogP measure that can describe more precisely the lipophilic feature of a drug dissolved in the bloodstream (pH 7.4) rather than in gastric acids (pH 1–2). Nevertheless, the LogP value, partly for its simplicity, is still considered a precious indicator to take into account in pharmaceutical and environmental chemistry. As an example, one of the empirical rules that describes a potential bioactive drug, formulated in 1997 by Lipinski [148] and some successive updates, identifies a LogP value included in the −0.4 to 5 range as suitable for a drug to be useful pharmacokinetically useful. Moreover, the LogP value is abundantly used as a molecular descriptor in quantitative structure–activity relationship (QSAR) calculations for many purposes.

It has to be mentioned that LogP values have not been experimentally determined for any known compounds and, often, calculated values are reported. Because many algorithms are disposable, it is possible to observe a discrepancy between different sources; it is recommended to check several reliable databases to verify the likelihood of the retrieved LogP magnitude order, as well as to compare data obtained from the same mathematical approach. As an example, experimental LogP values of different PCs, reported by the Drugbank database and different references, are shown in Table 1 [55,149,150,151,152,153]. The experimental LogP values of three dietary polyphenols (quercetin, rutin, and resveratrol) were reported by Rastogi et al. [55], analyzed by reverse phase high-performance liquid chromatography (RP-HPLC). Liu [150] reported dissimilar LogP values of the same polyphenols from blueberries using the ChemBioDraw Ultra 11.0. In addition, differing LogP values of apigenin andcirsimartin from rosemary were similarly reported using the ChemDraw software1 Ultra version 8.0 [149] compared with the Drugbank database and other studies. From Table 2, it is also noticeable that the experimental LogP values reported for the considered PCs are in the 0.15 to 3.62 range, with a mean value of 1.99. The most hydrophilic appears to be chlorogenic acid (small molecules with a ionizable acid function), together with rutin, a glycoside coupled with the disaccharide rutinose. The aglycon of rutin, the flavonolquercetin, shows a dramatic rise in lipophilicity (1.7 points in log scale) as a consequence of the loss of two sugar units. All the reported flavonoids (quercetin, apigenin, cirsimaritin, and daidzein) show similar LogP values, while the most lipophilic compound in the list iscurcumin, characterized by aromatic rings conjugated with insaturations, higher non-polar surface, and few polar groups in their chemical structure compared with other considered PCs. Regarding BCS permeability classes, the recommendation set by the United States Food and Drug Administration (USFDA) guide was to consider an absorption value of ≥90% as highly permeable [154]. This was subsequently revised, and the value was lowered to 85%, according to the World Health Organization (WHO) guide [155]. Metoprolol is commonly used as a reference [151], because its gastrointestinal membrane permeability lies in this boundary range. For this reason, Metoprolol LogP (1.8 according to Drugbank.com, accessed on April 2021) was used as a boundary marker in order to classify compounds as highly permeable (LogP value ≥ 1.8) or scarcely permeable (LogP value < 1.8). Relaying on this convention, ferulic acid, chlorogenic acid, rutin, and ellagic acid are considered scarcely permeable, while quercetin, apigenin, cirsimaritin, daidzein, resveratrol, and curcumin are highly permeable.

Regarding the LogP value, it may be considered as a helpful instrument to investigate drugs’ bioavailability, but effective absorption and metabolism mechanism are much more complex, and specific enzyme and sites interaction also play a relevant role. As an example, it was shown that emodin and chrysophanol exhibited different in vitro activities compared with apigenin and resveratrol [156]. This was attributable to differences in chemical structures, which have different affinities for drug-metabolizing enzymes and transporters, and P-gp, which are found in the intestinal epithelial cells. In particular, it was found that resveratrol absorption involves cis and trans isomers, which are characterized by both a different chemical stability and a differing interaction with intestinal transporters. Based on these properties, resveratrol is highly permeable. Furthermore, it was seen that the free hydroxyl groups of apigenin and resveratrol cause very rapid glucuronidation and sulfation reactions (phase II drug metabolism, Figure 2, step 5) in the liver or in the intestine, which leads to the production of the same quantity of glucuronidated and sulphated metabolites, respectively. Conversely, chrysophanol and emodine are present in the plasma mainly in the form of glucuronidated metabolites, compared with sulfonated metabolites [93]. A feasible explanation is that the methyl groups present in chrysophanol and emodine impede the production of the sulphated metabolite, and thus increase the possibility of interaction between the hydroxyl groups and the glucuronide enzyme.

### 4.2. Solubility

Solubility (S) is the concentration of a substance at equilibrium in a saturated solution in the presence of excess undissolved solid. In other words, solubility of a solute is the maximum quantity that can be dissolved in a certain quantity of solvent or solution at a specified temperature [157]. Considering that water solubility allows a drug an easy way of subministration, the diffusion capability in aqueous environments of the body, and the attainment of a desired drug concentration in systemic circulation, it clearly promotes the therapeutic activity of a specific compound [158]. Solubility of a given compound in a solvent is influenced by so many parameters that an exhaustive dissertation of the argument is outside the of this review; nevertheless, it is worthwhile to mention that S is dependent on pH, temperature, pressure, polarity of solvent and solute, ionization, enthalpy (solid phases and dissolution), H-bond formation, complex formation, and so on. Solubility can be expressed by different units, involving mass or volume ratios between solute and solvent, but the most used and officially established are concentrations units such as molarity [mol L^−1^] (moles of solute per liter of solvent) or mass/volume ratio [g L^−1^] (grams of solute per liter of solvent, and multiples). The magnitude of water solubility ranges widely among all the known compounds, from totally soluble (without limit) such as ethanol, miscible with water in every proportion, to nearly insoluble, such as silver chloride. A number of other descriptive terms have been developed to qualify the extent of solubility in particular fields. For example, U.S. Pharmacopoeia qualifies substances in terms of mass part of solvents required to dissolve one mass part of solute: very soluble (<1), freely soluble (between 1 and 10), soluble (between 10 and 30), sparingly soluble (between 30 and 100), slightly soluble (between 100 and 1000), very slightly soluble (between 1000 and 10,000), and insoluble (>10,000) [158]. BCS classification concerning solubility involves dividing substances into two classes: highly soluble or lowly soluble. Class boundaries are defined by the capability of the highest single immediate-release dose to dissolve into a specific volume of water (derived from typical bioequivalence study protocols, which prescribe administration of a drug product to fasting volunteers with a glass of water). A drug is considered highly soluble when the highest dose strength is soluble in 250 mL of water over a pH range from 1 to 6.8 (7.5 according to the FDA) [65] at 37 °C. In general, the regulatory authorities consider an active pharmaceutical ingredient (API) highly soluble if its highest dose (D) on solubility ratio (D/S) is less than 250 mL. The former European Medicines Agency guideline (EMA, formerly EMEA, 2001) and the present U.S. FDA (2000) guidance define dose as the highest dosage strength marketed as an oral immediate release (IR) dosage form; that is, the tablet or capsule with the highest content of API. However, the revised EMA (2010) guideline defines dose as the highest single oral IR dose recommended for administration in the Summary of Product Characteristics (also known as the Prescribers’ Information). The WHO has a more flexible definition. If the API appears on the WHO Model List of Essential Medicines (EML), the highest dose recommended in that list is to be applied for D/S ratio calculation. For APIs not on the EML, the highest dosage strength available on the market as an oral solid dosage form is used [159]. Clearly, the concept of “the highest dosage strength marketed as an oral IR dosage form” cannot be applied to dietary active principles. With the aim of referring to an analogous value, a sort of “single dietary abundant dose” concept is now proposed and used to determine PC solubility classification. The richest dietary source and the average content value is identified in the literature for any active principle; in the case of PC, www.Phenol-explorer.eu (accessed on 28th April 2021) is particularly useful. For any food categories (i.e., fruits, spices, vegetables, and so on), a weight value marked as “dietary abundant dose” is defined by analyzing consumption statistical data: it is the average daily consumption of an aliment from the highest consuming region of the world. As an example, in the case of fruits, Dominican Republic was found to be the highest consuming region, with an average of 125 kg per year per capita, equivalent to 342.5 g a day [160]. Similarly, representative food portionz have been determined for chocolate (31.5 g), spices (4.05 g), and processed soy (150 g). These values, multiplied for the average food content for any substance, are used as highest “single oral IR dose” or “single dietary abundant dose” (SDAD). Using the same criterion, SDAD values have been determined for any considered PC. In Table 2, water solubility, richest source, and D/S ratio (calculated as described) values are reported. According to the D/S ratio obtained from the reported assumptions, ferulic acid, chlorogenic acid, rutin, quercetin, apigenin, and cirsimaritin are classified as highly soluble, while daidzein, resveratrol, ellagic acid, and curcumin are considered scarcely soluble, showing a value greater than 250 mL.

SDAD is an artifact proposed in order to make up for the absence of established dose values for dietary active principles. Moreover, Fong et al. [151] proposed a similar approach to extend BCS classification to Chinese herbal traditional medicine. Moreover, SDAD calculations are revealed to be more consistent than a different strategy considered to solve the following question: referring to acceptable daily intake (ADI) values, calculated on the “no observed adverse effect level”(NOAEL), which is the higher concentration subministered that reveals no toxic effect on organisms, it can possibly give a representative measure of a “high dosage”. Nevertheless, PCs demonstrated a low toxicity and, consequentely, high ADI values that automatically would lead to cataloging any PC as unsoluble.

## 5. BCS Classification of Polyphenols

The BCS was first introduced for the development of new drugs as an aid to guide drug discovery [57]. Based on this system, both permeability and solubility are the two main factors influencing the level of drug transport through the cells of the intestinal mucosa, as well as the relationship between the concentration and the degree of absorption of the respective drug [149,154,164,165,166,167]. According to Papich [168], the solubility of a compound within biological compartments, as well as its permeability through biological membranes, limit both the degree and the extent of oral absorption of a drug.

The classification of specific chemical compounds using the BCS system permits an objective evaluation of their bioavailability for the human organism. According to the BCS, drugs can be classified into four categories: Class I (high solubility, high permeability), Class II (low solubility, high permeability), Class III (high solubility, low permeability), and Class IV (low solubility, low permeability). Regarding the four different BCS categories, drugs belonging to Class I have a high rate of both absorption and solubility. Hence, Class I drugs rapidly undergo first-pass metabolism, resulting in reduced concentrations that are then not bioavailable to the organism. Class II drugs are readily absorbed, but have a low solubility rate, which limits the in vivo absorption of the substance, except when administered at high doses. However, the absorption of these drugs is slower than Class I drugs and, therefore, they act over a longer period of time. For Class III drugs, the low permeability results in a limited absorption into the body. As these drugs are highly soluble, they are rapidly dissolved and, therefore, their activity is not dependent on the dose. Finally, Class IV drugs, having both low solubility and permeability, are problematic for effective oral administration, owing to their limited bioavailability [155,165,169,170,171]. Although the BCS was originally developed for orally administered drugs, it may also prove to be useful for bioactive substances contained in food, such as PC. In the following Table 3, the information concerning the BCS of PCs examined in this review is gathered, divided between those collected from the current literature and those based on the solubility and permeability criteria evaluated during this work.

Examining the current literature on this topic, PCs are generally characterized by a low solubility and high permeability, i.e., they belong mainly to BCS Classes II and IV, representing a limitation to their potential health benefits [162]. In addition, no PCs assigned to BCS Class I have been found. As shown in Table 3, resveratrol is considered as belonging to BSC Class II, as it is characterized by low water solubility and high lipophilicity [175]. In fact, resveratrol showed a rapid phase II drug metabolism within enterocytes and hepatocytes, owing to the fact that the -OH groups are engaged in labile acetal linkages [178]. Because of this, the bioavailability of resveratrol is limited and, consequently, its health benefits [110].

Similar to resveratrol, apigenin was also indicated as belonging to BCS Class II [174], whereas rutin, FA, and chlorogenic acid were shown to belong to BCS Class III [151]. The metabolism of these compounds was shown to be influenced by cellular absorption, requiring active transport mechanisms to be effectively absorbed by enterocytes [179]. In vitro studies revealed that FA is rapidly absorbed, but the excretion rate is low, and only 64% of the glucuronide metabolite is metabolized. Likewise, long retention times for FA in vivo were recorded, thus its activity time is prolonged in the human body [180]. Furthermore, according to Perez-Sanchez et al. [149], cirsimaritin was also shown to belong to BCS Class III.

Regarding curcumin, other authors placed it in Class IV of BCS classification because of its poor solubility and intestinal permeability. In fact, it is shown that low plasma and tissue levels associated with curcumin administration are due to its poor aqueous solubility, poor absorption, rapid metabolism, and rapid systemic elimination [177]. Daidzein is a PC commonly assigned to BCS Class IV [181]. Because of its low solubility and poor bioavailability, daidzein may have limited pharmaceutical value. It is further reported that daidzein shows a weak acidity and low permeability [152]. Finally, as reported by Nyamba et al. [176], ellagic acid seems to belong to BCS Class IV.

Controversies have been found in the literature regarding quercetin. In particular, it was shown to belong to either BCS Class I, II, or IV by Fong [151], Madaan [172], and Waldmann [173], respectively. As reported by different studies, this could be attributed to the type of dietary food from which the compound was extracted and analyzed, thereby influencing the bioavailability and biopharmaceutical classification.

Following the solubility criteria purposed by this review, and according to the definitions of high and low permeability and solubility explained in the previous paragraphs, it was possible to elaborate a new BCS classification for the compounds examined in this work. It emerged that polyphenols such as quercetin, apigenin, and cirismaritin belong to BCS Class I, as they are characterized by both a high LogP value (permeability) and high solubility. Quercetin and apigenin fully display the features of this class, showing both fast absorption and excretion, while cirsimaritin behaves more like a BCS Class III drug. It is relevant to mention that, in the case of quercetin, the BCS classification is questionable, given the fact that the available LogP values (reported in Table 1) lie around the edge of the reference parameter (1.8). Regarding BCS Class II, daidzein, resveratrol, and curcumin were identified as belonging to this class, as they were shown to have a high permeability, but poor solubility. Daidzein and resveratrol demonstrate coherent behavior, showing fast absorption and slow excretion, resulting in a low peak concentration, but long retention time within the organism; contrarily, curcumin shows scarce bioavailability owing to low permeability. Despite the LogP value, curcumin permeability is affected by interactions with gastrointestinal mucous membrane proteins, acting as a barrier. Displaying low permeability and high solubility, FA, chlorogenic acid, and rutin are assigned to BCS Class III. FA shows fast metabolism and excretion rate and non-dose-dependent peak plasmatic concentration, as expected; class III substances, on the other hand, demonstrate fast absorption, which may be explained by diffusion through enterocytes tight junctions. Chlorogenic acid also confirms the BCS Class III characteristic of fast excretion paths. Rutin has been assigned to BCS Class III, but its elevated hydrophilicity represents a limit for absorption, resulting in low bioavailability Finally, only ellagic acid, characterized by low permeability and solubility, is placed in BCS Class IV, well in agreement with in vivo data, stating low bioavailability and the fast reaching of a plateau in systemic concentration.

In conclusion, comparing the data obtained from the criteria proposed by this review with those gathered from the literature, there is good correspondence for FA, chlorogenic acid, rutin, resveratrol, and ellagic acid. Differently, the compounds highlighted in orange in Table 3 (apigenin, cirsimaritin, daidzein, and curcumin) are found to belong to a different class than those reported by the literature. This may be explained by the fact that the LogP values of these PCs are obtained through different databases, each of which uses different algorithms, leading to obtaining values of LogP (and thus permeability) that differ from each other. Moreover, referring to the SDAD parameter purposed by this review may lead to doses values sensibly different from those considered in other works.

## 6. Conclusions

Polyphenols have been widely studied for their several health benefits, and they also are used and examined from nutraceutical point of view, such as supplements. From this perspective, their distribution within the human body and their classification in pharmacological terms lead the interest of this review on the BCS classification of this compounds. Moreover, existing controversies in the literature regarding their classification further increase the interest of this review in evaluating and clarifying the currently available data.

Approaching the BCS classification on dietary bioactive principles, such as PC, immediately raises the problem that there is no univocal definition of maximum strength dose for these compounds. This leads to confusion because each author adopts an arbitrary criterion, or reports data from other studies without considering their context and even omitting necessary information. This is accompanied by variability in LogP values, as they are reported by the various databases as being calculated by different algorithms rather than experimentally verified. All of this results in an indetermination of the boundaries that should define the four classes provided by the BCS system. It would be recommended to identify univocal criteria, which may allow to define the administrable dose of dietary bioactive compounds, consequently allowing to establish, through unambiguous methods, the BCS class for these compounds as well, which are mainly assumed through diet. Nevertheless, the classification proposed by this review largely overlaps with the data found in previous literature, whereas controversies and mismatches largely arise from the definition of the criteria adopted to establish the administrable dosage.

It should be kept in mind that the BCS classification was created to study the behavior of active principles purposely subministred exclusively for therapeutic use, while dietary PCs, although manifesting bioactivity on human health, are taken without a purely pharmaceutical purpose.

However, for the 10 PCs examined, the BCS classification seems to be consistent with the bioavailability data found in the current literature. Checking PCs attributed to BCS Class I, quercetin and apigenin fully display the features of this class, showing both fast absorption and excretion, while cirsimaritin behaves more like a BCS Class III drug (LogP value may possibly be unreliable). Concerning BCS Class II attribution, daidzein and resveratrol demonstrate coherent behavior, showing fast absorption and slow excretion, resulting in low peak concentration, but long retention time within the organism; contrarily, curcumin shows scarce bioavailability owing to low permeability. Despite the LogP value, curcumin permeability is affected by interactions with gastrointestinal mucous membrane proteins, acting as a barrier. Ferulic acid shows fast metabolism and excretion rate and non-dose-dependent peak plasmatic concentration, as expected; class III substances, on the other hand, demonstrate fast absorption, which may be explained by diffusion through enterocytes tight junctions. Chlorogenic acid also displays the BCS Class III characteristic of fast excretion paths. Rutin has been assigned to BCS Class III, but its elevated hydrophilicity represents a limit for absorption, resulting in low bioavailability. Only ellagic acid has been assigned to BCS Class IV, well in agreement with in vivo data, stating low bioavailability and the fast reaching of a plateau in systemic concentration.

## Figures and Tables

**Figure 1 ijms-22-05514-f001:**
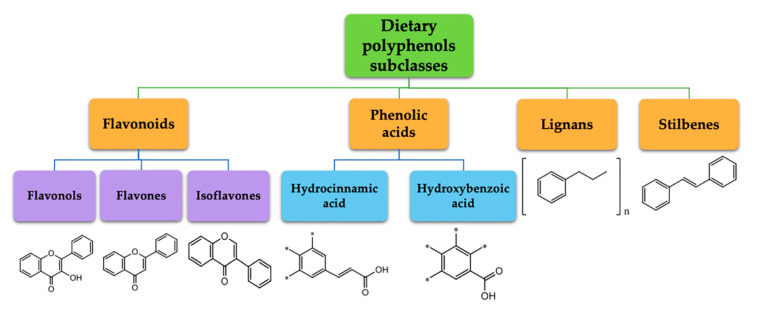
Dietary polyphenols known subclasses scheme.

**Figure 2 ijms-22-05514-f002:**
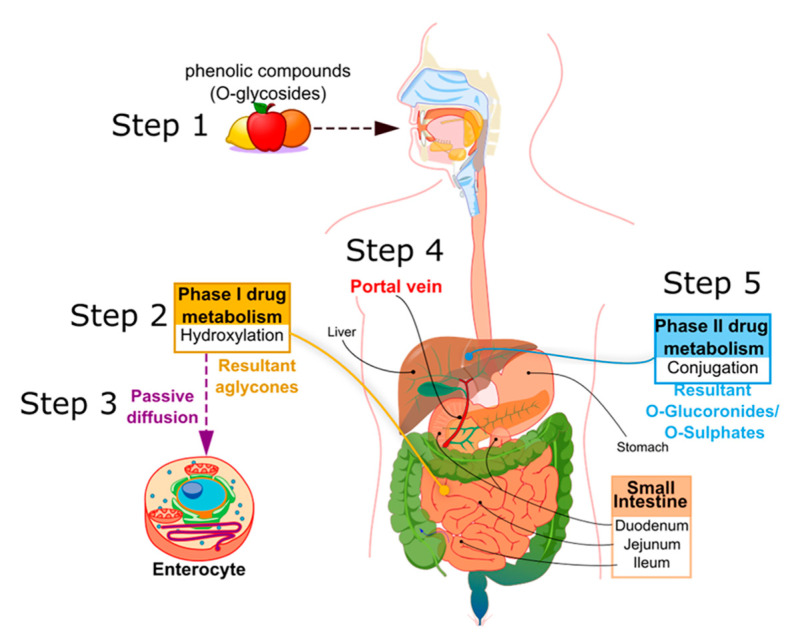
Principle metabolic steps of dietary phenolic compound (PC) in humans.

**Table 1 ijms-22-05514-t001:** Experimental LogP values reported by Drugbank.com database compared with miscellaneous references.

Compound	Structure	Drugbank LogP	Other Sources LogP
***Ferulic acid***	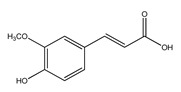	1.58	1.42 [150]
			0.96 [151]
***Chlorogenic acid***	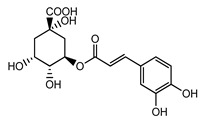	0.17	0.37 [151]
			−0.3 [153]
***Rutin***	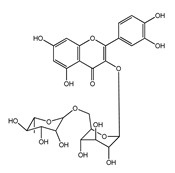	0.15	0.47 ± 0.01 [55]
			−2.28 [150]
			−0.9 [151]
***Quercetin***	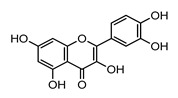	1.81	1.59 ± 0.06 [55]
			0.35 [150]
			1.82 ± 0.32 [152]
			1.48 [153]
***Apigenin***	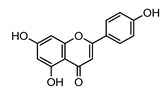	3.07	1.51 [150]
			1.9 [149]
			2.92 ± 0.06 [152]
***Cirsimaritin***	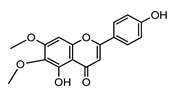	N/A	2.04 [149]
***Daidzein***	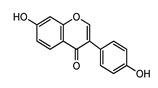	3.3	2.13 [150]
			2.51 ± 0.06 [152]
***Resveratrol***	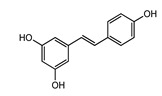	2.57	2.99 ± 0.13 [55]
			3.06 [150]
			3.1 [153]
***Ellagic acid***	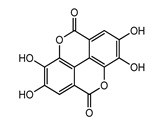	1.59	1.366 http://www.chemspider.com (accessed on 28th April 2021)
***Curcumin***	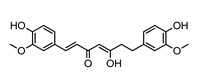	3.62	3.29 [153]

**Table 2 ijms-22-05514-t002:** Water solubility, sources, calculated single dietary abundant dose (SDAD), and calculated dose on solubility ratio (D/S) values of principle phenolic compounds.

Compound	Experimental Water Solubility	Richest Source	SDAD	D/S
***Ferulic acid***	0.78 mg mL^−1^ http://www.chemspider.com (accessed on 28th April 2021)	Dark chocolate (240 mg kg^−1^)	7.56 mg	9.69 mL
***Chlorogenic acid***	40 mg mL^−1^ [153]	Plums (758.8 mg kg^−1^)	259.9 mg	6.49 mL
***Rutin***	0.13 mg mL^−1^ [161]	Capers (3323 mg kg^−1^)	13.45 mg	103.46 mL
***Quercetin***	0.0003 mg mL^−1^ [162]	Dark chocolate (250 mg kg^−1^)	7.87 mg	26.23 mL
***Apigenin***	0.00216 mg mL^−1^ [162]	Origanum majorana (44 mg kg^−1^)	178.2 µg	82.5 mL
***Cirsimaritin***	2.57 × 10^−4^ mg mL^−1^ [149]	Origanum vulgare (684.3 mg kg^−1^)	2.77 mg	10.78 mL
***Daidzein***	0.008215 mg mL^−1^ [162]	Tempeh (processed soy) 136 mg kg^−1^	20.4 mg	2483.2 mL
***Resveratrol***	0.03 mg mL^−1^ [162]	Cranberry, red (30 mg kg^−1^)	10.27 mg	342.3 mL
***Ellagic acid***	0.0093 mg mL^−1^ [162] 0.0097 mg mL^−1^ [163]	Chestnuts (7354 mg kg^−1^)	2.518 g	270,752 mL
***Curcumin***	11 × 10^−6^ mg mL^−1^ [162]	Curry powder (2853 mg kg^−1^)	11.55 mg	10,500,000 mL

**Table 3 ijms-22-05514-t003:** Biopharmaceutical classification system (BCS) classification of phenolic compounds (PCs) based on the solubility and permeability criteria evaluated in this review and based on the current literature.

Family	Compounds	BCS Classification	
		Reviewed	Literature
**Hydroxycinnamic acid**	Ferulic acid	III	III [151]
	Chlorogenic acid	III	III [151]
**Flavonols**	Rutin	III	III [151]
	Quercetin	I	I [151] or II [172] or IV [173]
**Flavones**	Apigenin	I	II [174]
	Cirsimaritin	I	III [149]
**Isoflavones**	Daidzein	II	IV [175]
**Stilbenes**	Resveratrol	II	II [176]
**Tannins**	Ellagic acid	IV	IV [177]
**Curcuminoids**	Curcumin	II	IV [178]

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
