# Peer review of "An Overview on Dietary Polyphenols and Their Biopharmaceutical Classification System (BCS)"

_ijms, 2021, doi:10.3390/ijms22115514_

Round 1
Reviewer 1 Report
The manuscript is very well written and organized and easy to understand.
Minor revisions should be made in order to be published in IJMS journal, and the manuscript should be completed and/or modified taking into account few suggestions:
- The authors are advised to rephrase the sentences from lines: 261-263, 277-279, 301-302, 540-542, 550-551.
- The authors are advised to correct the words: Metroprolol (line 379), have (line 584).
- The authors are advised to change the conclusions, there are too many details.
Author Response
Thank you for the kind suggestions. The paper has been modified according to them and the changes have been highlighted. In particular, the following corrections have been done:
- The authors are advised to rephrase the sentences from lines: 261-263, 277-279, 301-302, 540-542, 550-551.
All the sentences have been rephrased accordingly (highlighted in yellow)
- The authors are advised to correct the words: Metroprolol (line 379), have (line 584).
The words “Metroprolol” and “have” were corrected (highlighted in yellow).
3.The authors are advised to change the conclusions, there are too many details.
The conclusions have been changed according to the reviewer’s suggestion. In particular, the lines 591-606 were deleted from the conclusion paragraph and some sentences were added in “BCS Classification of Polyphenols” paragraph, in particular lines 540-542, 546-550, 551-556, 557-558 were added (highlighted in blue)

Reviewer 2 Report
The suggestions previously made by the reviewer were considered in the preparation of this revised version of the manuscript. No further comments.
Author Response
Thank you for revising the manuscript and for accepting the changes that the authors provided accordingly.
This manuscript is a resubmission of an earlier submission. The following is a list of the peer review reports and author responses from that submission.
Round 1
Reviewer 1 Report
This comprehensive study reviews the bioavailability of different phenolic compounds in terms of intestinal absorption, solubility, and classification according to the biopharmaceutics classification system (BCS), thus providing an overview of the information regarding the LogP values of polyphenols, and modified metabolic by-products, and their association with the BCS classification. This is an innovative and relevant study within the research field in which it fits, bringing up a subject that has been little explored and discussed by the scientific community. This review may help to clarify some issues frequently discussed by authors who work with health-promoting effects of phenolic compounds. In addition, the manuscript is well written and structured and is easy to read. I suggest the publication of this study after a minor revision.
Additional comments/suggestions:
- Although a list of abbreviations has been provided, the names should be given in full the first time they appear in the text of the manuscript.
- In the title, BSC may be written in full.
- It is suggested that the chemical structure of some of the phenolic compounds discussed in this review be presented in additional figure.
Reviewer 2 Report
I think that the subject is interesting. However, the manuscript seems to be covered so many items and details. I would recommend you to focus on specific components or subclasses of interests. Any abbreviations used should be defined at first mention throughout the manuscript. For more detailed information see the Information for Authors.
Through the manuscript, you said that it will provide an overview of the information regarding the LogP values of polyphenols, and modified metabolic byproducts, and their association with the BCS classification. Many information was provided, but it is hard to get it easily.
Polyphenol is such a too broad and summative concept. There are so many subclass or individual components, and yet so many of them are still unknown. I think it would not be appropriate to use this term because you only covered several individual components. And I’m not sure that you can sum up their availability in one section. I recommend you to reorganize by adding subtitle on each component. Or you need to be more focused on specific subclasses. If you’d like to maintain “polyphenol” concept, I think you should define this more specifically in the introduction section: for example, on how many various component you will cover, etc. For polyphenol metabolism, I think you’d better to mention only “metabolism” related things. If you’d like to mention about bottleneck or limitation, I think you should separate them from metabolism part.
In abstract, it was mentioned that “These compounds are widely consumed in the Mediterranean diet and many 10 studies report several benefits on human health thanks to their bioavailability in humans”. But these compounds are present in various plant-based foods. It cannot be limited in such one certain thing. And it was said that PC will benefit not only for drug, but also for food. It wasn’t specified how.
Reviewer 3 Report
The manuscript entitled “An overview of polyphenols and their BCS classification” presents a review on metabolism, bioavailability, as well as the BCS classification of polyphenolic compounds.
Although the paper is interesting, there are some serious questions regarding the novelty and the significance of content of the manuscript.
- Numerous studies already investigated the absorption and bioavailability of polyphenols.
- The polyphenolic compounds are a large group of at least 10,000 different compounds. The authors present only some information of very few compounds (max 15), therefore the title does not clearly reflect the content of the manuscript. For instance, in Table 1, for 7 compounds „No literature info” for Quantity ingested from food and bioavailability, in terms of concentration in plasma and urinary excretion.
- The authors should explain first before using some abbreviations (for instance: PC, N/A, etc.)
- The authors should correct – from line 227, 229: „hemodine”; lines 235-236: „HPLC liquid chromatography method”
Therefore, although I appreciate the authors hard work, I do not think that the manuscript is suitable for publication in IJMS journal.
Round 2
Reviewer 2 Report
Through the abstract, LogP seems to be an important factor, but not solubility. If you want to express the importance of LogP, I think you should rephrase more based on the sentences between line 300 and 311. If this is really important concept than solubility, it would be better make/rephrase a paragraph consisting of the definition, their relation, how to apply this concept for food research area or future research on polyphenols. If both LogP and LogS are important, it should be mentioned equally.
Abstract: Based on the information in the Tables, it would be better to add how many polyphenols were reviewed in this manuscript and their names.
Line 20: If objective of this review to find the potential of BCS as a method for food-related research area, Biopharmaceutical needs to be changed to a term like Nutraceutical.
Line 33‒42: Dietary polyphenol subclasses the authors mentioned are well-known examples or well-reported examples. So current classification is based on what we’ve known, but not entirely discovered. It would be better to be careful for the wording.
Ling 57‒58: Consequently, the health benefits of PC are limited due to what reasons? For clarification, it would be better to mention again for which specific reasons to limit the health benefit.
Line 76: Please describe full name for SGLT1.
Line 91: Edit the space between the sentences.
Line 115: in vitro should be set in italics.
Line 132: Remove “However”. It doens’t seem necessary to put.
Line 182: FA seems the abbreviation of ferulic acid. According to the Author’s guideline, any abbreviations used should be defined at first mention throughout the manuscript, even though they were mentioned in the list of abbreviations.
Line 130‒218: Throughout “3. Polyphenol Bioavailability”, If you’d like to expand the meaning of these components and their availability, I would suggest to add some epidemiological results which linked dietary consumption of this and their metabolites in blood or urine samples of general population. These intervention studies are definitely helpful, but adding these evidence will make more understandable. Would it be possible to suggest or show their maximum or mean abrosption rate through this reference review?
Line 244‒245: “It was shown that emodin and chrysophanol exhibited different in vitro activities compared to apigenin and resveratrol.[8896]” What is the reason to bring up emodin and chrysophanol? Because they weren’t mentioned before, I don’t get it why these brought suddenly, even if their differences compared with apigenin and resveratrol were brought.
It seemed that more detailed information or calculation methods or detailed equation on the calculation of LogP may be more helpful for understading. But software, mentioned in Line 259-260 seems not meaningful to mention.
Line 284: Figure 3 didn’t seem to be meaningful in this article. It can be mentioned in the text, such as “Under the definition of solubility by Kansara”. But is there any official definition of solubility?
Line 344‒355: It seems drug specific information doens’t need to be explained. If the authors want to, it would be better to explain to link more about food related components. It doesn’t need to be put the BCS classification. It seems more appropriated to mention that “Under the available classification, PC seemed to be characterized by a low solubility and high permeability,”
Line 373‒379: What components are linked to cirsimaritin, hispidulin, and scutellarein? If these are connected to other PCs in Table 2 or 3. In Table 3, cirsimaritin and hispidulin had no water solublity. What is the meaning of putting these two componets in Table 3? And Scutellarein was not mentioned. It would be better to put contents more
Line 398: Is it really that daidzein should be classified in BCS class II? Can it be that certain? It may need to tone down.
Line 399 and 401. Detailed explanation on Class 1‒4 in the Figure 4 can be added on the Table 4 and 5. But is it necessary to divide into two tables? It seems that presenting Table 5 seems sufficient.
Ling 404‒424: What is the exact conclusion? The needs for developing new classification system? Or Regardless of its limitation, BCS classification still be useful for PC?
Reviewer 3 Report
Although the paper is interesting and the authors made some corrections, still there are some serious questions regarding the novelty and the significance of content of the manuscript.
- Numerous studies already investigated the absorption and bioavailability of polyphenols.
- The phenolic compounds are a large group of at least 10,000 different compounds. The authors present only some information about very few polyphenolic compounds (max 15), therefore the title does not clearly reflect the content of the manuscript, even if the title was changed.
Therefore, although I appreciate the authors hard work, I do not think that the manuscript does not meet the publishing requirements by IJMS.